# Understanding Relationships between Cultivated Land Pressure and Economic Development Level across Spatiotemporal Characteristics: Implications for Supporting Land-Use Management Decisions

**DOI:** 10.3390/ijerph192316362

**Published:** 2022-12-06

**Authors:** Dan Yang, Zhenyue Liu, Pengyan Zhang, Zhuo Chen, Yinghui Chang, Qianxu Wang, Xinyue Zhang, Rong Lu, Mengfan Li, Guangrui Xing, Guanghui Li

**Affiliations:** 1College of Geography and Environmental Science, Henan University, Kaifeng 475004, China; 2Key Laboratory of Geospatial Technology for the Middle and Lower Yellow River Regions, Henan University, Ministry of Education, Kaifeng 475004, China; 3Regional Planning and Development Center, Henan University, Kaifeng 475004, China; 4Department of Geography, Kent State University, Kent, OH 44242, USA

**Keywords:** cultivated land pressure (CLP), economic development, coupling relationship, land-use management, affected areas of the lower Yellow River (AALYR)

## Abstract

Food security is crucial to world peace. Economic development has posed a great threat to the protection of cultivated land. Considering 20 cities in the lower Yellow River (AALYR) as the study area, this study explored the spatial evolution of cultivated land pressure (CLP) and economic development from 1998 to 2018, revealing the spatiotemporal coupling characteristics of the CLP index and economic development. The main results are as follows: we discerned that CLP and economic development have an obvious spatiotemporal consistency during 1998–2018. The CLP showed a spatial pattern of overall stability, as well as local changes. Most prefecture-level cities experienced decreased significantly in CLP and improvements in food security. Overall, there were regional differences in the coupling relationships between CLP and economic development in the study area. The explanatory power of the proportion of secondary and tertiary industries were significantly higher than other driving factors. Therefore, while developing the economy rapidly, we should also protect cultivated land resources and improve the coordination level between them, which is essential to guarantee food security and a steady economic development.

## 1. Introduction

Food security relates to the overall national economic and social development situation and is an important cornerstone in ensuring national security [1]. In April 2020, the World Food Program released a research report showing that the number of people facing severe food insecurity would increase, implying that the global food security situation was not optimistic. As a major grain producer, China’s food security has always attracted significant attention [2,3]. Lester R. Brown, president of the Worldwatch Institute in the United States, said that China’s shortage of cultivated land would cause a large shortfall in relation to food demand, thus directly threatening world food security. Cultivated land is a typical multi-functional natural resource and its use and supply directly affect the development of the economy. Simultaneously, cultivated land resources are continuously decreasing due to rapid economic development [4]. As such, the pressure on cultivated land is too high, which directly affects food security [5]. With the advancement of urbanization in China, cities with dense populations and more developed economies are facing the problem of serious decline in the quality and the area of cultivated land [6]. Some trends of cultivated land conversion are irreversible, putting great pressure on resources [7]. Therefore, studying the internal mechanisms of the relationship between regional CLP and economic development has great practical significance for ensuring regional food security and a sustainable economic development.

Western scholars have studied cultivated land resources since the 1930s [8,9]. Due to the varying amounts of cultivated land in different countries, the research focus on the pressure brought by the conversion of cultivated land has also varied significantly. Studies on cultivated land resources worldwide mainly concern land utilization rate and intensification, agricultural ecological structure and industrial occupation of cultivated land [10]. Ramankutty et al. [11] combined satellite data with national agricultural data to create a 10-km spatial resolution dataset and analyzed the spatial changes in global agriculture. Baessle [12] pointed out that the excessive intensive utilization of cultivated land resources would impact regional ecological environments. Jaiyeoba [13] focused on the effects of the changes in the intensive use of cultivated land on soil properties. Compared with European and American countries, Asian countries (e.g., Japan and South Korea) are in short supply of cultivated land resources. Coupled with rapid economic development, the issue of occupied cultivated land area is more serious and the relationship between man and land is becoming more acute. The literature mainly focuses on research on cultivated land conversion and cultivated land protection, as well as the regulatory measures taken by the government to protect cultivated land [14]. After comparing the increase with the decrease in industrial and agricultural land in South Korea, Korean scholars believe that a powerful method to prevent cultivated land conversion is to control land use [15], while Japanese scholars believe that agricultural land should be subject to special control through regulation laws for agricultural revitalization areas and the formulation of agricultural land laws [16]. Globally, many scholars have hitherto paid significant attention to the international issues of changes in cultivated land use and food security while developing their own food security solutions. For example, in the 1990s, foreign scholars raised the question, “Who will feed China?” based on a study of food security in China [17].

Although China has diversified research perspectives on CLP, the core starting point is food security [18]. The CLP index based on the interrelationship between food production and demand has become the preferred method for scholars to evaluate the level of regional CLP; thus, it is widely used in regional food security evaluations [19]. This method comprehensively considers the relationship between population, cultivated land and food. However, previous studies have mainly focused on cultivated land resources’ carrying capacity [20], the relationship between cultivated land protection and urbanization [21], food security [22,23], spatial difference in cultivated land quality [24], driving factors [25], dynamic response of cultivated land use change [26] and the correlation effect of the economic development mode on cultivated land use change [27]. Additionally, scholars have discussed the possibility of the dynamic balance of total cultivated land [28], the multi-functional transformation of cultivated land [29] and the impact of the transformation of cultivated land on food production [19,30] and on ecosystem service value [31].

In terms of the analysis of the spatial differences in CLP in different regions, scholars have adopted the kernel density estimation method [32], system dynamics model [33], center of gravity model [34] and spatial autocorrelation [35], based on different spatial scales, to study China’s cultivated land resources, mainly manifested by differences among the eastern, central and western areas [36], along with interprovincial [37] and inter-city differences [38]. These results laid a foundation for the scientific understanding of the spatiotemporal characteristics of regional CLP and have high academic value. The spatial and temporal variation of CLP among regions is the result of multiple factors. The existing research mostly focuses on natural and grain production factors [25,39] such as natural disasters, climate change, agricultural inputs and food production costs and prices, paying little attention to economic development factors.

In general, the existing research has systematically described the evolutionary law of the spatiotemporal patterns of CLP, laying a foundation for understanding the utilization of cultivated land and regional agricultural development. However, most studies use a single visual method to observe the spatiotemporal differences between regions and pay less attention to the coupling relationship between CLP and economic development within a region.

The Yellow River basin is regarded as the key area in relation to achieving food security in China [40,41]. The food output in this area accounts for 13.4% of China’s total food output, thus playing an extremely important strategic role in ensuring national food security [42]. As the birthplace of China’s agricultural civilization, the AALYR has a large population density, resulting in a small per capita cultivated land area. Additionally, the continuous expansion of the urban construction scale has further reduced the cultivated land area [43]. Therefore, protecting the existing cultivated land resources while developing the economy has become the key to ensuring food security and sustainable economic development in the AALYR. Based on this, this study uses ArcGIS 10.3 spatial analysis, the CLP index, the spatial measurement method and geographic detector model to explore the spatial correlation, coupling law and driving factors of CLP and the economic development level in the AALYR from 1998 to 2018. It thus provides scientific support for the formulation of regional agricultural development policies, rational use of land and consensus regarding the contradiction between economic development and cultivated land use.

## 2. Study Area

As the birthplace of China’s agricultural civilization, and also a major food-producing areas, the AALYR are mainly composed of plains and hills that have a warm climate and rich natural resources [44]. The central and northern plain areas have flat terrain and fertile soil and have made important contributions to the food production of Henan and Shandong, playing a pivotal role in ensuring national food security and national economic construction. The AALYR include the Henan and Shandong provinces (Figure 1). As of 2018, the population in this region was 119.98 million and the total food output was 74.986 million tons.

## 3. Methods and Data Sources

### 3.1. Methods

#### 3.1.1. Evaluation of CLP Index

(1)The minimum per capita cultivated land area (MPCA) refers to the cultivated land area that can meet people’s normal food needs under the regional food productivity level and supply capacity, which takes into account the linkage between cultivated land and other factors [45]. In any country, changes in cultivated land are inevitable with industrialization and urbanization. When the actual per capita cultivated land area (ACPA) in a region is lower than the MPCA, it will endanger basic food security. The formula is:

(1)$$S_{\min}=\frac{\beta \cdot G_{r}\cdot M}{Z}$$ where *S*_min_ is the MPCA (hm^2^/person) and *β* is the food self-sufficiency rate (%). Since the AALYR is the main food production area in China, this study considers the self-sufficiency rate of food to be 100%. *G_r_* is the per capita food demand (kg/person), *M* is the total cultivated land area (hm^2^) and *Z* is the total food output (kg).

(2)CLP index is a variable that varies according to time and space. It is usually used to measure the tension between regional cultivated land resources. The CLP index can provide a benchmark for scientific regulation of regional cultivated land resources [33]. The formula is:

(2)$$K_{j}=\frac{S_{\min}}{S_{a}}$$ where *K_j_* is the CLP index and *S_a_* is the APCA (hm^2^/person). The value of *K_j_* reflects the pressure level of cultivated land resources at a specific time and place. When *K_j_* = 1, the value of cultivated land resources protection indicates that the regional food output meets the demand of people; when *K_j_* < 1, the CLP is not significant and regional cultivated land resources are abundant; when *K_j_* > 1, the regional cultivated land resources are under great pressure and corresponding measures should be taken to ensure an effective supply of food to meet people’s consumption needs.

#### 3.1.2. Geographic Correlation Rate and the Center of Gravity Model

The geographical correlation rate is an indicator that reflects the connection between two economic factors in their geographical distribution. This method can be used to better describe the correlation between the CLP index and the economy [46]. The calculation model is:(3)$$G=100-\frac{1}{2}\sum_{i=1}^{n}\left|S_{i}-P_{i}\right|$$
where *G* is the geographical connection rate, *S_i_* is the weight of the contribution of the CLP of each city to the CLP of the entire region and *P_i_* is the ratio of the economic development level of each city to that of the entire region. When the geographical distributions of *S_i_* and *P_i_* are relatively consistent, the value of *G* is larger, indicating a higher rate of geographical connection between the two elements; conversely, when the geographical distribution difference between *S_i_* and *P_i_* is larger, the value of *G* is smaller, indicating that the geographical relationship between these two elements is not too tight.

To further reveal the coupling relationship between the CLP index and economic development, the barycenter theory is introduced. The barycenter of a certain attribute point in an administrative region can be characterized by certain attributes and geographical coordinates of a sub-administrative region [47].
(4)$$X_{t}=\frac{\sum_{i=1}^{n}M_{i}X_{i}}{\sum_{i=1}^{n}M_{i}},\;Y_{t}=\frac{\sum_{i=1}^{n}M_{i}Y_{i}}{\sum_{i=1}^{n}M_{i}}$$
where *M_i_* is the CLP index or economic development level of each city and (*X_t_*, *Y_t_*) is the regional CLP index or economic development level center of gravity in year *t*.

To reflect the gravity center offset, we introduce the gravity center space offset distance *d*:(5)$$d=111\cdot \sqrt{{\left(X_{tj}-X_{ti}\right)}^{2}+{\left(Y_{tj}-Y_{ti}\right)}^{2}}$$
where (*X_ti_*, *Y_ti_*) and (*X_tj_*, *Y_tj_*) are the *i*th and *j*th barycenter coordinates in year *t* and *d* is the spatial distance between the two barycentric coordinates.

#### 3.1.3. Geographical Concentration and Coupling Index

Geographical concentration is an indicator of the concentration of certain attributes in a region, which can be used to measure the spatial distribution of specific elements [48], The formula is:(6)$$R_{i}=(A_{i}/\sum_{i=1}^{n}A_{i})/(T_{i}/\sum_{i=1}^{n}T_{i})$$
where *R_i_* is the CLP index or geographical concentration index of the economic development level in region *i*, *A_i_* is the CLP index or economic development level in region *i* and *T_i_* is the land area of region *i*.

The spatial coupling relationship between CLP and economic development can be expressed by the ratio of the geographical concentration of the two. The formula is [49]:(7)$$I_{i}=\frac{R_{i\;CLP}}{R_{i\;economic\;development}}$$
where *I_i_* is the coupling index of region *i*, *R_i CLP_* is the geographical concentration of CLP in area *i* and *R_i economic development_* is the geographical concentration of the economic development level of region *i*.

#### 3.1.4. Geographical Detector Model

The geographic detector is a new method used to reveal the driving force behind the spatially stratified heterogeneity of dependent variables [50]. It has become an appropriate approach for the geographical exploration of regional and differential mechanisms. The *q* statistic value is the interpretation degree of independent variables factor to dependent variable. The range is (0, 1) and the value represents the influence of related factors on the coupling degree of CLP and economic development. The formula is:(8)$$q=1-\frac{1}{N\sigma^{2}}\sum_{s=1}^{E}N_{s}\sigma_{s}^{2}$$
where *q* is the explanation degree of the spatial differentiation of the coupling degree between CLP and economic development, *N_s_* is the number of sample units in the secondary regions, *N* is the number of sample units in the entire area, *E* is the number of secondary regions, *σ^2^* is the variance of coupling degree and *σ_s_^2^* is the variance of secondary regions.

### 3.2. Data Sources

Considering reliability and availability, agricultural and economic data on 20 cities in the AALYR from 1998 to 2018 were mainly collected from provincial and prefecture-level statistical yearbooks. These include *Henan Statistical Yearbook* (1999–2019), *Shandong Statistical Yearbook* (1999–2019) and *China Urban Statistical Yearbook* (1999–2019). Data missing from the above yearbooks were supplemented by those from the *Statistical Bulletin on National Economic and Social Development*. Digital elevation model data were obtained from the Resources and Environmental Sciences and Data Center, Chinese Academy of Sciences (http://www.resdc.cn, accessed on 1 March 2022) and the coordinates of urban administrative centers were used as urban geographic coordinates.

## 4. Results

### 4.1. Changes in CLP Index in the AALYR

With the acceleration of urbanization, the CLP will inevitably increase and the CLP index can reflect the pressure level of regional resources [51]. Many scholars in China set the value of the annual per capita food demand at 300–400 kg/person, while The World Food and Agriculture Organization proposed that 400 kg/person can meet a normal nutritional demand. Therefore, the per capita food consumption demand from 1998 to 2018 was set as 400 kg/person in this study [52]. The two provinces in the AALYR are China’s major agricultural and food provinces. The food self-sufficiency rate at 100% is relatively high [49]. The calculation results are shown in Figure 2.

#### 4.1.1. Temporal Variation Characteristics of CLP

We obtained the trend of arable pressure in the AALYR region during 1998–2018 based on Equations (1) and (2). As can be seen from Figure 2, MPCA and CLP showed an “M” and inverted “V” decreasing trend respectively, while the overall fluctuation of APCA was more moderate. Among them, the APCA in the AALYR was relatively stable from 1998 to 2003 and the APCA was highest in 2014; further, the MPCA increased from 0.030 hm^2^ in 1998 to 0.074 hm^2^ in 2003 and the CLP index showed an increasing trend. In 2003, the CLP was highest. After 2003, the MPCA and the CLP index entered a stable decline path.

#### 4.1.2. Spatial Variation Characteristics of CLP

To further explore the similarities and differences of the spatial distribution of CLP in 20 cities in the AALYR, based on the principle of time interval consistency, we calculated the CLP index of the cities in the AALYR in 1998, 2002, 2006, 2010, 2014 and 2018 according to the CLP index model and divided the CLP index into five groups [53]; the thresholds were set as 0.9, 1.1, 1.5 and 2. We used ArcGIS10.3 software to visualize the CLP index in the AALYR in space (Figure 3). When the CLP index is below 0.9, it indicates that the cultivated land area is not under pressure (absolute food security) and 0.9–1.1 is the warning pressure area (critical point of food security); the urban food security in the above two areas is better. Specifically, a CLP index of 1.1–1.5 represents a low CLP area (slight food risk area), 1.5–2 indicates a medium CLP area (food risk area) and a value greater than 2 shows a high CLP risk area (serious food risk area).

From Figure 3, it can be seen that the CLP in the AALYR presents a spatial distribution pattern of “overall stability and local changes”. From 1998 to 2018, the CLP in most prefecture-level cities in the AALYR was at the no pressure level and only a few prefecture-level cities were above the low-pressure level. The CLP was significantly reduced and the food security was effectively guaranteed. In 1998, the CLP level in the AALYR was low; from 2002 to 2006, the CLP changed significantly due to the low-temperature freezing and drought disasters; from 2006 to 2010, the CLP level in Jinan, Jining, Dongying and Zibo showed a downward trend; from 2010 to 2018, except for Zhengzhou, Laiwu and Zibo, the CLP in other cities was at the level of alarm pressure or no pressure.

From the perspective of regional distribution, the cities with no CLP in the AALYR were mainly distributed in the southwest. This is because Henan, an important food production area, is rich in cultivated land resources. Although Jinan, as the provincial capital city, is indirectly affected by the surrounding cities, the APCA is constantly increasing. Due to the increase in food production per unit area and the increase of grain sowing proportion, the growth of the CLP index is slowed and the CLP increased slightly. From 1998 to 2018, Zhengzhou and Laiwu experienced increased pressure on cultivated land. As the provincial capital city, Zhengzhou’s rapid urbanization process and the increase in the non-agricultural population pose a threat to cultivated land, which were the main factors for the excessive CLP. Especially, the development orientation of the transportation hub and central city meant that economic development was pursued at the expense of the protection of cultivated land. Laiwu is located in the hinterland of central Shandong Province. The urbanization rate increased rapidly from 31% in 1998 to 67% in 2018. The amount of built-up land was increasing greatly, occupying cultivated land, causing a continuous decrease in cultivated land, and the urban CLP was gradually increasing. The alarm pressure level in 1998 has changed to a high-pressure level in 2018, indicating that the current development situation has posed a threat to the cultivated land resources and it is urgent to protect the cultivated land.

### 4.2. Characteristics of Economic Development Changes

As a major agricultural region in China, the AALYR grow not only grain crops such as wheat, corn and soybeans, but also economic crops such as cotton, oil and tubers. According to statistical data (Figure 4), the economy in the AALYR has developed rapidly in recent years. In 1998, the GDP was 616.638 billion yuan and the per capita GDP was 6181.5 yuan/person. In 2018, the GDP reached 6877.69 billion yuan and the per capita GDP was 57,300 yuan/person. From 1998 to 2018, the GDP growth rate in the AALYR was 10.5% and there was a large difference in economic scale among cities. In 2018, the smallest regional GDP was held by Hebi with a value of 86.19 billion yuan, while the largest, Zhengzhou had a value of 1014.33 billion yuan, 11.77 times that of Hebi. In 2018, the per capita GDP of Dongying was the largest, with 211,000 yuan, while that of Zhoukou was the smallest, with 21,000 yuan. The per capita GDP of Dongying was about 10 times that of Zhoukou. By the end of 2018, the primary industry in the AALYR accounted for 7.3% of the area’s GDP and the secondary industry accounted for 46.1% of GDP. With the continuous improvement of economic development levels, the entertainment and service industries were booming, exceeding the proportion of the secondary industry. In 2018, the tertiary industry accounted for 46.6% of GDP; therefore, the secondary and tertiary industries were dominant.

### 4.3. Geographical Relationship and the Evolution of the Center of Gravity for CLP and Economic Development

The geographical connection rate can effectively measure the spatial distribution relationship between CLP and the economy in the cities in the AALYR from 1998 to 2018. As can be seen from Figure 5, the geographic connection rates of GDP and per capita GDP in YYLAR regions were greater than 87 (the highest value being 96) and showed a decreasing trend, with GDP decreasing at a faster rate. This implied that the trends of GDP and per capita GDP were basically consistent. Overall, there was still a large variability in their trends at different stages during the study period.

To further reveal the spatio-temporal dynamics between CLP and GDP and per capita GDP, we quantified and obtained a high spatio-temporal trend of coupling between CLP and GDP based on Equations (4) and (5). Figure 6 shows that the path of CLP moves from southwest to northeast and then shifts to the southwest. Namely, it moved 0.30° in the meridional direction and 0.15° in the zonal direction, a phenomenon that indicates the unstable change of cultivated land. From 1998 to 2018, the movement trend of the center of gravity of the CLP index and GDP in the AALYR was generally consistent, showing an inverted U-shaped trend. From 1998 to 2002, the center of gravity of the CLP, GDP and per capita GDP moved to the northeast, the moving distance of the center of gravity of the CLP index being large at 17.47 km. The center of gravity of the CLP was always located in Heze, while the center of gravity of the GDP was transferred from Heze to Jining. After 2006, the center of gravity of the GDP moved back from Jining to Heze, with a moving distance of 13.46 km. The center of gravity of the per capita GDP was located in Tai’an, with a moving distance of 5.70 km. In general, the CLP and the moving direction and trajectory of the GDP’s center of gravity in the AALYR are consistent, which indicates that the regional economic development level has a certain correlation with the CLP index.

### 4.4. Spatial Differentiation of the Coupling between CLP and Economic Development

#### 4.4.1. Spatial Coupling Pattern of CLP and Economy

To further reveal the coupling relationship between the CLP and economic development, the geographical concentration and coupling index of CLP, GDP and per capita GDP of 20 cities in the AALYR were calculated by Formula (6). We found geographical variability in the coupling relationship between cultivated land pressure and economic development. Table 1 shows that, from 1998 to 2018, according to the geographical concentration of CLP, peak areas appeared in Laiwu, Hebi, Jiaozuo, Zhengzhou and Zibo. The low-value areas were Dezhou, Zhoukou and Shangqiu. From the perspective of economic geographical concentration, high-value areas appeared in Laiwu, Zibo, Jiaozuo and Zhengzhou and the gap between the highest and the lowest value was significant, indicating significant differences in regional economic development. Regarding the coupling index of CLP and the economy, the indexes of Laiwu, Hebi and Zhengzhou were higher, indicating that the growth of regional CLP had a closer relationship with economic growth in recent years and that economic development affected the level of CLP. The coupling indexes of Jinan and Dongying were low and the CLP lagged behind their economic development, indicating that the growth rate of CLP brought by the economic growth of the two cities was lower than that of other cities.

#### 4.4.2. Spatial Coupling Types of CLP and Economy

According to the matching relationship between CLP and economic development in the AALYR, calculate the coupling indexes of CLP and GDP, per capita GDP and take its average value to represent the coupling index of CLP and economic development. According to Figure 7, the 20 cities in the AALYR are divided into three types according to their coupling indexes: 0 < I ≤ 1 represents lagging CLP, 1 < I ≤ 1.5 is the coordinating type of CLP and economic development and I ≥ 1.5 the leading type of CLP [54].

From the perspective of the spatial coupling type, most regions of the AALYR showed a decoupling trend between economic growth and CLP. Among them, in 1998, there were eight cities with CLP lagging behind economic development, namely Dezhou, Tai’an, Liaocheng, Jining, Xinxiang, Zhengzhou, Zhoukou and Shangqiu; here, the impact of economic development on changes in CLP was small. From 1998 to 2014, these cities changed from the lagging type to the coupling and coordination types. By 2018, the number of cities of this type had increased, indicating that the regional pressure on cultivated land was low, the level of economic development was high and that economic development was less affected by the CLP. However, the development of the cities coordinated by CLP and economic development was relatively stable, with a high degree of coupling between the two, showing a good interactive relationship. From 1998 to 2014, this type of city was concentrated and located in Henan Province, indicating that the CLP and economic development were gradually balanced in this province. By 2018, the number of cities with coupling coordination between CLP and economic development decreased from nine in 2014 to seven, most of which were located in the southwest. The number of prefecture-level cities whose CLP was ahead of economic development increased from three in 1998 to six in 2010. By 2018, only Zhengzhou, Laiwu and Hebi were ahead of the economic development cities. Overall, the impact of regional economic growth on the CLP has been gradually decreasing, as is the number of cities with leading pressure on cultivated land. However, the CLP and economic development in some cities show a relatively high coupling degree. In the long run, it is thus necessary to further strengthen the construction of high standard cultivated land, actively develop the ecological agriculture, improve the level of food output and prevent problems such as an insufficient food supply and unreasonable use of cultivated land resources caused by the excessive CLP.

### 4.5. Driving Factors of the Coupling Degree between CLP and Economic Development

In this study, the coupling degree influencing factor model was constructed using seven indicators, namely the standard cultivated land coefficient (X1), fertilizer use per unit area (X2), proportion of irrigation area (X3), proportion of secondary and tertiary industries (X4), total mechanical power per unit of grain sown area (X5), pesticide use per unit of grain sown area (X6) and rural per capita disposable income (X7). Among them, the standard cultivated land coefficient is the standard measure of the quality of cultivated land. Fertilizer use per unit area, the proportion of irrigation area, total mechanical power per unit grain sown area and pesticide use per unit grain sown area represent the input level of agricultural production factors and facilities. The secondary and tertiary industries have a direct impact on the employment structure of farmers. As such, the rural per capita disposable income determines the input ability of farmers to grow grain and also has a certain impact on their enthusiasm to do so.

Based on the indicators constructed above, this study used a natural breakpoint method to divide the coupling degree of CLP and economic development in the AALYR into five levels; the *q* values of each factor of the coupling degree of CLP and economic development in the AALYR are obtained by the geographical detector method (see Table 2). From the determinant *q* value of each index factor, there are great variations among different influencing factors and the proportion of the secondary and tertiary industries is the largest, which is significantly different from the other factors. This shows that economic factors can significantly promote the rational development of the coupling between the CLP and economic development, which is the dominant driving factor. The secondary and tertiary industries have a high employment rate, which has a direct impact on the employment structure of farmers and can provide a broad resettlement space for the non-agricultural employment of agricultural labor, thus hindering the improvement of food production efficiency and resulting in greater pressure on regional cultivated land. The standard cultivated land coefficient, rural per capita disposable income, fertilizer use per unit area and pesticide use per unit area are affected by their properties and calculation methods, which have little explanatory power but have an important impact on the coupling development of the two factors, thus being general level driving factors. The explanatory power of the irrigated area proportion and the total power of unit agricultural machinery are weak and have no significant effects. In general, being affected by the unbalanced development of various cities, the explanatory power and the degree of influence of various factors are quite different. The explanatory power of the proportion of secondary and tertiary industries is strong, while the degree of influence on the coupled development of CLP and economic level is large. In the future, while developing the economy, we should strengthen the optimization and adjustment of the industrial structure and, simultaneously, increase the comprehensive impact level of various factors, so as to improve the coordinated development level of CLP and economic growth.

## 5. Discussion

As a non-renewable resource, cultivated land is the basis of human production and life, being part of a complex system of social services, agricultural production, ecological protection and other functions [55]. With the continuous increase in the agricultural production level, science and technology level and population, the depth and breadth of land resource utilization have been strengthened et the food demand, causing a sharp increase in the pressure on the cultivated land resource carrying capacity system [56]. Consequently, exploring the relationship between economic development and CLP is of practical significance for regional land resource utilization and agricultural policy formulation.

With the continuous advancement of urbanization, the speed of land use change has been accelerated and a large ratio of cultivated land is being used for other land use types. From 1998 to 2018, the CLP in the AALYR showed a change trend in its “overall stability and local aggravation.” In cities with rapid economic development, such as Zhengzhou, the CLP showed an increasing trend, which proved that the increase in CLP can’t be separated from the economic development of regional cities [57,58]. Over time, the CLP was relatively high in some years, reaching its highest value in 2003 [59]. Being mainly affected by flood disasters, the cultivated land was submerged and the food output decreased significantly, which increased the pressure on the cultivated land. After 2003, the MPCA and CLP entered a steady declining path. On the one hand, the land system benefited from the continuous improvement of agricultural production technology in China [60] and, on the other hand, the government improved the land system, strictly examined and approved land use and reduced the occupation of cultivated land [61]. Therefore, the above two aspects have effectively alleviated the CLP in the AALYR.

In terms of spatial characteristics, sufficient agricultural development, a high grain seed ratio and food production in the AALYR were observed. After 2010, the CLP in other cities, except for Zhengzhou, Laiwu and Zibo, was measured as a grade of alarm pressure or no pressure [62]. Simultaneously, there was a high coupling relationship between the CLP and economic development over time and space, which indicates that the increase in farmers’ income brought by economic development is closely related to the pressure on cultivated land. Compared with the focus on economic development, the shift of the center of gravity of CLP is more sensitive, mainly because the CLP is affected by regional policies, population change, regional urbanization and other factors [63]. From 1998 to 2018, the CLP and the coordinated cities decreased in terms of economic development. As a major agricultural province, Henan Province showed a high degree of consistency in the coupling between the CLP and economy; further, the CLP and the speed of economic development were relatively matching. By 2018, Zhengzhou, Laiwu and Hebi were the leading cities in terms of CLP, which indicates that economic growth is more closely related to CLP and the growth of the CLP caused by economic growth is common [64]. This is due to rapid industrial development, which causes cultivated land to be occupied, leading to an obvious impact of economic growth on the cultivated land [5,65]. Therefore, coordinate the relationship between CLP and economic development is the key to reduce the CLP and achieve sustainable development.

Current studies on the relationship between CLP and economic development mostly focus on spatiotemporal distribution characteristics, with less research on the drivers of the coupling between the two. Based on this, this study further investigates the drivers of the coupling relationship between CLP and economic development in the AALYR by using the geographical detector. The results show that the degree of influence of each factor on the coupling relationship varies due to unbalanced development among cities. Among them, economic factors such as the proportion of secondary and tertiary industries have a significant influence on the change in the relationship between them; the input level of agricultural production factors and facilities, as well as the input ability of farmers to grow grain, also have important impacts [66,67]. Therefore, to prevented increasing pressure on cultivated land, measures such as regional land use control, basic cultivated land protection and overall land use planning should be implemented [5] to avoid the negative impact of the disorderly expansion of agriculture on cultivated land in the process of urbanization. This can subsequently improve production capacity, thus ensuring national food security.

The main contribution of this study is to discuss and analyze the geographic and coupling relationship between CLP and economic development in the AALYR region. In addition, it explains the drivers of changes in the coupling relationship through a geo-graphic detector model. However, the changes in the relationship between the two are still governed by various control variables, for example, disturbances in the natural environment, such as climate, topography and hydrology, as well as population migration, are all factors that lead to changes in their relationship. Future research is needed to reveal the influence of these potential factors on the relationship between the two, as well as to broaden the scale of research and reveal how the relationship between the two changes with spatial scale, so as to provide a basis for regional formulation of differentiated conservation and development policies.

## 6. Conclusions

The AALYR area has undergone a change in development concept from “big development” to “big protection” and its special function positioning has determined the double pressure of development and protection. Our study aims to explore the coupling relationship between CLP and economic development, delineate the types of spatial coupling and reveal the driving factors, which are important for the rational use of land resources, guaranteeing food security and stable economic development. Our study found that the CLP of the AALYR region showed a fluctuating downward trend during 1998–2018 and its relationship with economic development gradually shifted from a highly coupled to a decoupled type. In terms of moving trajectory, CLP tends to move in the northeast direction for both GDP and GDP per capita; in terms of influencing factors, the share of secondary and tertiary industries had the strongest explanation.

In general, the spatial correlation between CLP and economic development in AALYR is constantly changing. Although the trend between them is gradually changing for the better, there is still necessary to improve the quality of cultivated land, increase the level of food production and avoid wasting arable land. At the same time, we also suggest that, in the future, while developing the economy, the government should increase investment in agriculture to promote sustainable regional development and ensure food security.

## Figures and Tables

**Figure 1 ijerph-19-16362-f001:**
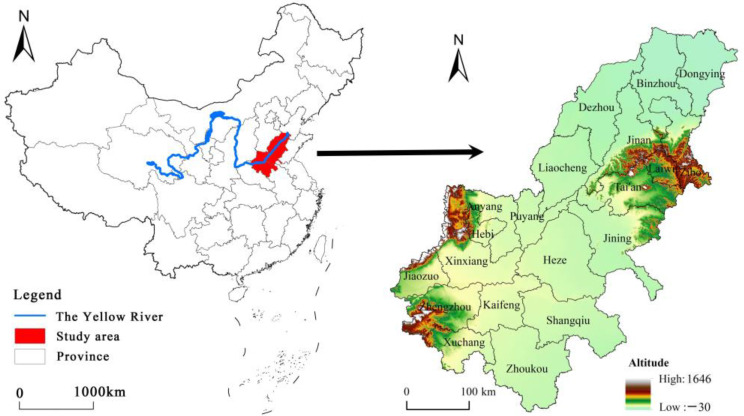
Study area.

**Figure 2 ijerph-19-16362-f002:**
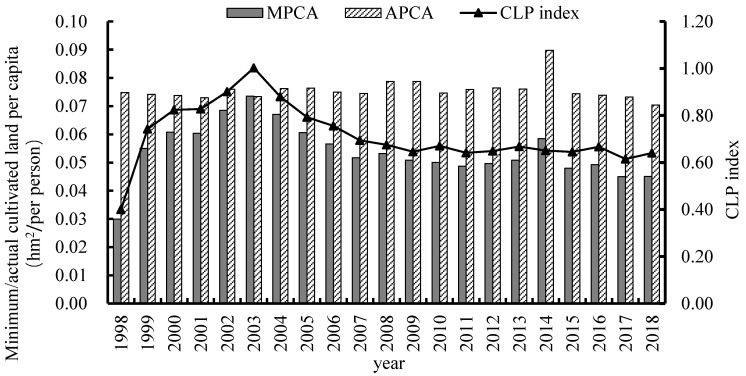
Change of MPCA, APCA and CLP index in the AALYR.

**Figure 3 ijerph-19-16362-f003:**
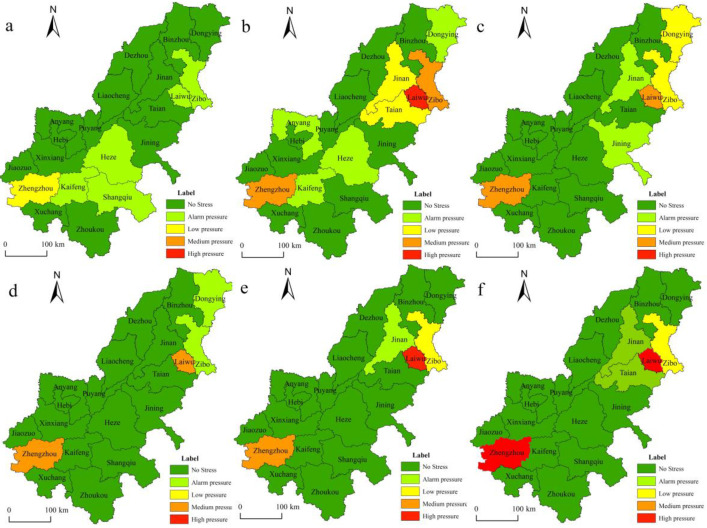
Spatial evolution of the CLP index. (Note: (**a**–**f**) in the figure represents 1998, 2002, 2006, 2010, 2014 and 2018, respectively.)

**Figure 4 ijerph-19-16362-f004:**
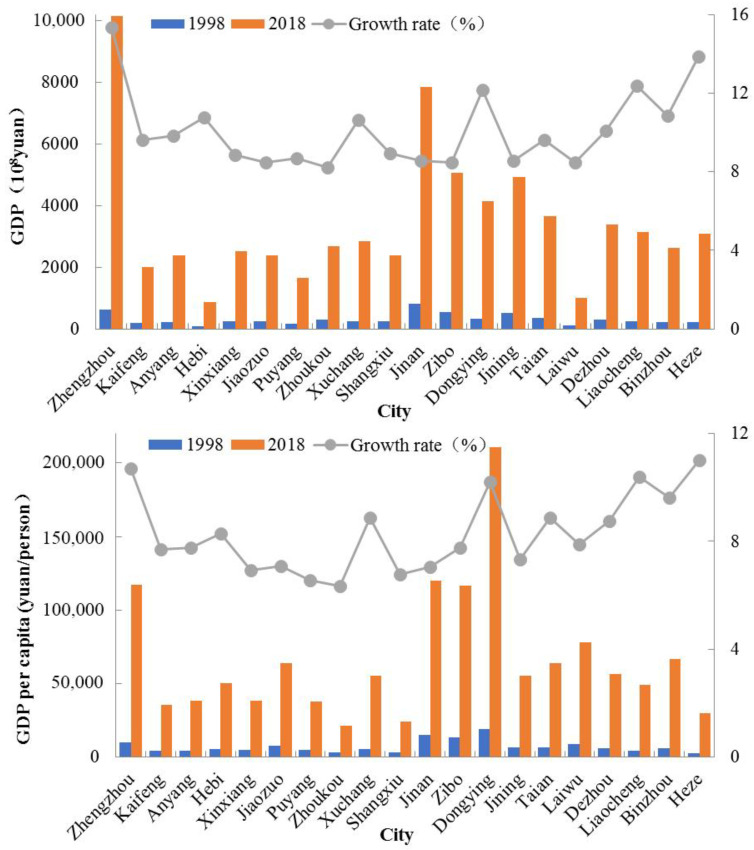
Economic development of the cities in the AALYR.

**Figure 5 ijerph-19-16362-f005:**
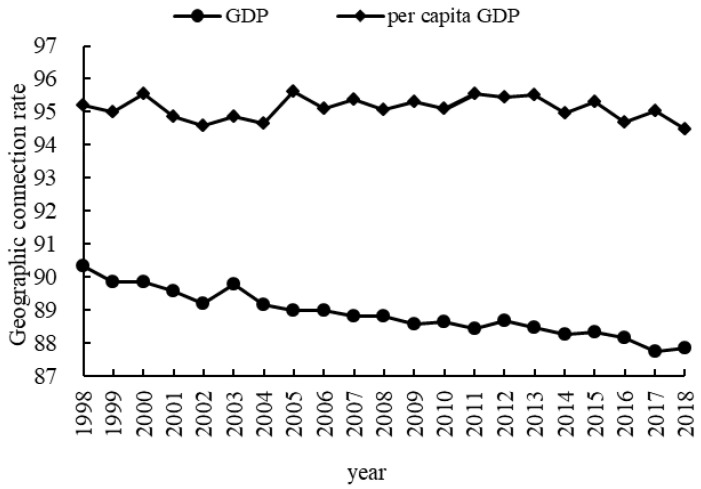
Change in the geographical connection rate between CLP index, GDP and per capita GDP.

**Figure 6 ijerph-19-16362-f006:**
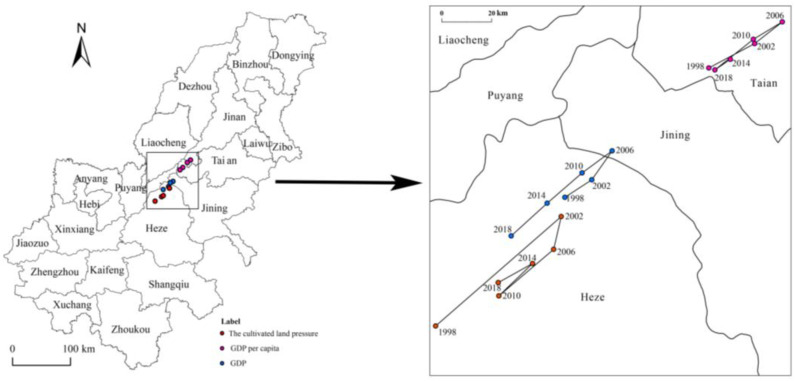
Center of gravity of the CLP index, GDP and per capita GDP in the AALYR.

**Figure 7 ijerph-19-16362-f007:**
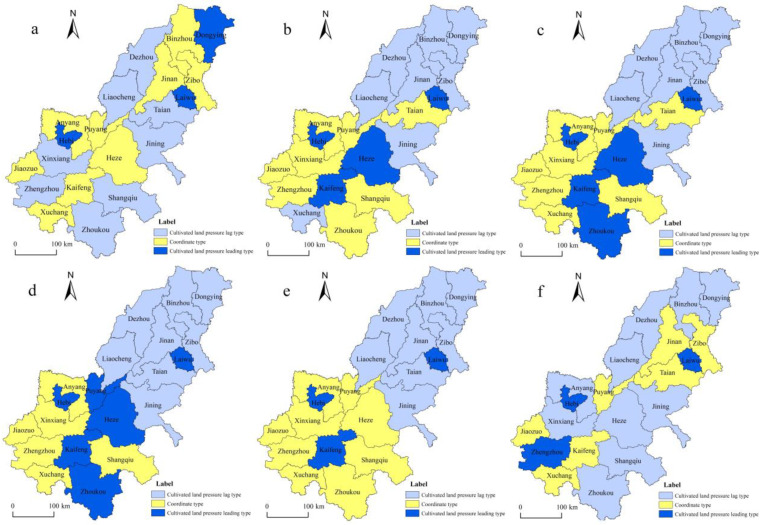
Coupling types of CLP and economic development in the AALYR. (Note: (**a**–**f**) in the figure represents 1998, 2002, 2006, 2010, 2014 and 2018, respectively.)

**Table 1 ijerph-19-16362-t001:** Geographical concentration and coupling indexes of CLP and economic development in the AALYR.

City	2002				2010				2018			
	R_1_	R_2_	R_3_	I	R_1_	R_2_	R_3_	I	R_1_	R_2_	R_3_	I
Zhengzhou	1.83	2.11	1.47	1.06	2.30	2.43	1.35	1.32	2.76	3.00	1.80	1.96
Kaifeng	1.04	0.71	0.67	1.51	1.24	0.64	0.65	1.93	1.12	0.71	0.65	1.14
Anyang	0.95	0.71	0.61	1.44	0.90	0.79	0.72	1.19	0.85	0.71	0.59	0.95
Hebi	2.42	0.81	2.57	1.96	2.55	0.88	2.75	1.91	2.48	0.87	2.64	1.86
Xinxiang	0.66	0.70	0.58	1.03	0.74	0.65	0.55	1.25	0.63	0.67	0.53	0.81
Jiaozuo	1.60	1.19	1.59	1.17	1.73	1.37	1.85	1.10	1.66	1.28	1.78	1.29
Puyang	1.20	0.94	1.19	1.14	1.46	0.81	1.07	1.58	1.36	0.87	1.04	1.21
Zhoukou	0.48	0.58	0.25	1.38	0.54	0.46	0.23	1.78	0.44	0.50	0.20	0.69
Xichang	1.20	1.23	1.23	0.97	1.36	1.18	1.29	1.11	1.28	1.25	1.27	1.14
Shangqiu	0.52	0.54	0.30	1.35	0.55	0.48	0.30	1.50	0.48	0.49	0.25	0.74
Jinan	1.10	2.48	1.94	0.50	0.98	2.14	1.49	0.56	1.22	2.17	1.71	1.37
Zibo	1.91	2.22	2.41	0.82	1.54	2.15	2.24	0.70	1.96	1.87	2.23	1.46
Dongying	0.86	1.15	2.95	0.52	1.12	1.33	3.09	0.60	0.61	1.11	2.92	0.83
Jining	0.61	1.18	0.66	0.72	0.61	1.00	0.58	0.83	0.63	0.97	0.56	0.81
Taian	1.12	1.12	0.92	1.11	0.88	1.18	1.01	0.81	1.12	1.04	0.94	1.06
Laiwu	7.66	1.07	3.85	4.58	7.77	1.09	3.97	4.54	8.43	0.99	3.96	4.76
Dezhou	0.49	0.75	0.62	0.72	0.30	0.72	0.61	0.46	0.30	0.72	0.62	0.57
Liaocheng	0.62	0.73	0.58	0.96	0.51	0.83	0.69	0.67	0.54	0.81	0.65	0.74
Binzhou	0.66	0.61	0.75	0.98	0.51	0.72	0.91	0.63	0.40	0.60	0.79	0.63
Heze	0.66	0.34	0.18	2.80	0.53	0.45	0.26	1.63	0.42	0.56	0.28	0.66

Note: R_1_, R_2_ and R_3_ are the geographical concentrations of CLP, GDP and per capita GDP, respectively, while I is the average value of the coupling index of CLP, GDP and per capita GDP, respectively.

**Table 2 ijerph-19-16362-t002:** Driving factors of the coupling degree between CLP and economic development.

Period	X1	X2	X3	X4	X5	X6	X7
1998–2002	0.181	0.265	0.154	0.348	0.173	0.217	0.169
2003–2006	0.232	0.140	0.130	0.188	0.214	0.135	0.166
2007–2010	0.227	0.186	0.203	0.170	0.184	0.113	0.165
2011–2014	0.177	0.199	0.152	0.214	0.146	0.169	0.165
2015–2018	0.181	0.266	0.154	0.349	0.173	0.218	0.169

## Data Availability

Not applicable.

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
