# Peer review of "Understanding Relationships between Cultivated Land Pressure and Economic Development Level across Spatiotemporal Characteristics: Implications for Supporting Land-Use Management Decisions"

_ijerph, 2022, doi:10.3390/ijerph192316362_

Round 1

Reviewer 1 Report

Comments

This is a very meaningful and successful article. From the perspective of food security, it analyzes and explores the relationship between cultivated land pressure and economic development by studying 20 cities in the lower reaches of the Yellow River. important. At the same time, I also have some small suggestions, which I hope will be helpful to your research.

1 In line 76, is "Who will feed China" changed to "Who will feed Chinese"?

2 In the results section, the author carefully analyzes the results of the research. Obviously, you have done a lot of work, but in each subsection, there is a lack of summary of the results of this part; for example: Section 4.1.1.

3 It is recommended to use box plots or bar charts to express the content of Table 1, which will make it easier for readers to understand.

4 It is recommended that the author reduce the number of words in the results section and use graphs to express the research results as much as possible;

5 The conclusion should be the main result of the article, and the author should present the research conclusion of the article, not the analysis; the author should carefully revise the content of the conclusion.

6 The authors are advised to add the understudy section.

Author Response

Point–by–point responses

Reviewer #1

This is a very meaningful and successful article. From the perspective of food security, it analyzes and explores the relationship between cultivated land pressure and economic development by studying 20 cities in the lower reaches of the Yellow River. important. At the same time, I also have some small suggestions, which I hope will be helpful to your research.

Comment 1: In line 76, is "Who will feed China" changed to "Who will feed Chinese"?

Response: Thank you for your comments. Since "Who will feed China" in line 76 of the article is from reference 17 at the end of the article, "17. Brown, L.R. Who Will Feed China? Wake-Up Call for a Small Planet. New York: WW Norton and Company, 1995.", we consider not to change the name of the original book, so we keep the same title as the source book in the article.

Comment 2: In the results section, the author carefully analyzes the results of the research. Obviously, you have done a lot of work, but in each subsection, there is a lack of summary of the results of this part; for example: Section 4.1.1.

Response: Thank you for your reminder. We have added the corresponding summaries in the corresponding sections as you suggested (corresponding to the first line of the revised document, respectively). The details are as follows.

Lines: 238-241:

“We obtained the trend of arable pressure in the AALYR region during 1998-2018 based on equations (1) and (2). As can be seen from Figure 2, MPCA and CLP showed an "M" and inverted "V" decreasing trend respectively, while the overall fluctuation of APCA was more moderate. Among them,”

Lines: 313-317:

As can be seen from Figure 5, the geographic connection rates of GDP and per capita GDP in YYLAR regions were greater than 87 (the highest value being 96) and shown a fluctuating decreasing trend, with GDP decreasing at a faster rate. This implied that the trends of GDP and per capita GDP were basically consistent. Overall, there was still a large variability in their trends at different stages during the study period.

Lines: 321-324:

To further reveal the spatio-temporal dynamics between CLP and GDP and per capita GDP, we quantified and obtained a high spatio-temporal trend of coupling between CLP and GDP based on equations (4) and (5).Figure 6 shows that the path of CLP moves from southwest to northeast and then shifts to the southwest.

Lines: 344-347:

We found geographical variability in the coupling relationship between cultivated land pressure and economic development. Table 1 shows that, from 1998 to 2018, according to the geographical concentration of CLP, peak areas appeared in Laiwu, Hebi, Jiaozuo, Zhengzhou, and Zibo.

Lines: 370-371:

From the perspective of the spatial coupling type, most regions of the AALYR shown a decoupling trend between economic growth and CLP. Among them,

Comment 3: It is recommended to use box plots or bar charts to express the content of Table 1, which will make it easier for readers to understand.

Response: Thank you for your comments. We have replaced the contents of Table 1 with Figure 4 according to your suggestion, and have revised the corresponding "Table 2" and "Table 3" in the original manuscript to "Table 1" and "Table 2" respectively, and "Figure 4", "Figure 5" and "Figure 6" in the original manuscript to "Figure 5", "Figure 6" and "Figure 7" respectively, in order to make the contents expressed in the article clearer and easier for readers to understand. The details are as follows.

Lines: 292:

“According to statistical data (Figure 4),”

Lines: 307-308:

Figure 4. Economic development of the cities in the AALYR.”

Lines: 319-320:

“Figure 5. Change in the geographical connection rate between CLP index, GDP, and per capita GDP.”

Lines: 339:

“Figure 6. Center of gravity of the CLP index, GDP, and per capita GDP in the AALYR.”

Lines: 358-359:

Table 1. Geographical concentration and coupling indexes of CLP and economic development in the AALYR.

City

2002

2010

2018

R1

R2

R3

I

R1

R2

R3

I

R1

R2

R3

   I

Zhengzhou

1.83

2.11

1.47

1.06

2.30

2.43

1.35

1.32

2.76

3.00

1.80

1.96

Kaifeng

1.04

0.71

0.67

1.51

1.24

0.64

0.65

1.93

1.12

0.71

0.65

1.14

Anyang

0.95

0.71

0.61

1.44

0.90

0.79

0.72

1.19

0.85

0.71

0.59

0.95

Hebi

2.42

0.81

2.57

1.96

2.55

0.88

2.75

1.91

2.48

0.87

2.64

1.86

Xinxiang

0.66

0.70

0.58

1.03

0.74

0.65

0.55

1.25

0.63

0.67

0.53

0.81

Jiaozuo

1.60

1.19

1.59

1.17

1.73

1.37

1.85

1.10

1.66

1.28

1.78

1.29

Puyang

1.20

0.94

1.19

1.14

1.46

0.81

1.07

1.58

1.36

0.87

1.04

1.21

Zhoukou

0.48

0.58

0.25

1.38

0.54

0.46

0.23

1.78

0.44

0.50

0.20

0.69

Xichang

1.20

1.23

1.23

0.97

1.36

1.18

1.29

1.11

1.28

1.25

1.27

1.14

Shangqiu

0.52

0.54

0.30

1.35

0.55

0.48

0.30

1.50

0.48

0.49

0.25

0.74

Jinan

1.10

2.48

1.94

0.50

0.98

2.14

1.49

0.56

1.22

2.17

1.71

1.37

Zibo

1.91

2.22

2.41

0.82

1.54

2.15

2.24

0.70

1.96

1.87

2.23

1.46

Dongying

0.86

1.15

2.95

0.52

1.12

1.33

3.09

0.60

0.61

1.11

2.92

0.83

Jining

0.61

1.18

0.66

0.72

0.61

1.00

0.58

0.83

0.63

0.97

0.56

0.81

Taian

1.12

1.12

0.92

1.11

0.88

1.18

1.01

0.81

1.12

1.04

0.94

1.06

Laiwu

7.66

1.07

3.85

4.58

7.77

1.09

3.97

4.54

8.43

0.99

3.96

4.76

Dezhou

0.49

0.75

0.62

0.72

0.30

0.72

0.61

0.46

0.30

0.72

0.62

0.57

Liaocheng

0.62

0.73

0.58

0.96

0.51

0.83

0.69

0.67

0.54

0.81

0.65

0.74

Binzhou

0.66

0.61

0.75

0.98

0.51

0.72

0.91

0.63

0.40

0.60

0.79

0.63

Heze

0.66

0.34

0.18

2.80

0.53

0.45

0.26

1.63

0.42

0.56

0.28

0.66

Lines: 367-368:

According to Figure 7, the 20 cities in the AALYR are divided into three types according to their coupling indexes:

Lines: 396:

“Figure 7. Coupling types of CLP and economic development in the AALYR.”

Lines: 414:

“(see Table 2).”

Lines: 437:

“Table 2. Driving factors of the coupling degree between CLP and economic development.”

Comment 4: It is recommended that the author reduce the number of words in the results section and use graphs to express the research results as much as possible.

Response: Thank you for your reminder. We have revised the results analysis section based on your suggestions to improve the accuracy and logic of the study results. The details of the changes are the same as comment 2 and 3 above.

Comment 5: The conclusion should be the main result of the article, and the author should present the research conclusion of the article, not the analysis; the author should carefully revise the content of the conclusion.

Response: Thank you for your comments. We have reorganized the conclusion section according to your suggestion (corresponding to lines 510-527 of the revised paper), summarized the relevant results of this study, further deepened the motivation of the study, and made the content studied in the article more understandable. The details are as follows.

 “The AALYR area has undergone a change in development concept from "big de-velopment" to "big protection", and its special function positioning has determined the double pressure of development and protection. Our study aims to explore the coupling relationship between CLP and economic development, delineate the types of spatial coupling, and reveal the driving factors, which are important for the rational use of land resources, guaranteeing food security and stable economic development. Our study found that: the CLP of the AALYR region showed a fluctuating downward trend during 1998-2018, and its relationship with economic development gradually shifted from a highly coupled to a decoupled type. In terms of moving trajectory, CLP tends to move in the northeast direction with both GDP and GDP per capita; in terms of influencing factors, the share of secondary and tertiary industries had the strongest explanation.

In general, the spatial correlation between CLP and economic development in AALYR is constantly changing. Although the trend between them is gradually changing for the better, there is still necessary to improve the quality of cultivated land, increase the level of food production, and avoid wasting arable land. At the same time, we also suggest that: in the future, while developing the economy, the government should increase investment in agriculture to promote sustainable regional development and ensure food security.”

Comment 6: The authors are advised to add the understudy section.

Response: Thank you for your comments. Our study is based on the reality of food security in China. With the rapid socio-economic development, cultivated land has gradually become the main contributor to urban expansion, and the area of "non-agriculturalized" cultivated land has gradually expanded. As a non-renewable resource, the relationship between cultivated land and economic development deserves our attention. Based on this, we conducted this study and found that the relationship between cultivated land pressure and economic development is gradually changing from a high coupling to a decoupling trend for the lower Yellow River influence area, but regional differences still exist. Our findings provide a reference for the development of differentiated cultivated land protection policies within the impact area. At the same time, there are certain limitations in our study, and has revised and supplemented in the discussion section (corresponding to lines 498-508 of the revision paper). The details are as follows.

 “The main contribution of this study is to discuss and analyze the geographic and coupling relationship between CLP and economic development in the AALYR region. In addition, it explains the drivers of changes in the coupling relationship through a geo-graphic detector model. However, the changes in the relationship between the two are still governed by various control variables, for example, disturbances in the natural en-vironment such as climate, topography, and hydrology, as well as population migra-tion, are all factors that lead to changes in their relationship. Future research is needed to reveal the influence of these potential factors on the relationship between the two, as well as to broaden the scale of research and reveal how the relationship between the two changes with spatial scale, so as to provide a basis for regional formulation of differenti-ated conservation and development policies.”

Reviewer 2 Report

Dear authors

your work is very interesting. For me it was a pleasure to review it. Only two comments you will find in the text:

a) I found it very difficult to understand the concept of "CPL". Also, I not understand the relationship with "K";

2) in the first part of the manuscript you used the unit "hm2". I do not know this unit and in the text I suggest to change it to "km2". Is this my gap??? 

Good luck

Author Response

Point–by–point responses

Reviewer #2

Your work is very interesting. For me it was a pleasure to review it. Only two comments you will find in the text:

Comment 1: I found it very difficult to understand the concept of "CPL". Also, I not understand the relationship with "K";

Response: Thank you for your reminder. We have changed the first occurrence of "CLP" to "cultivated land pressure (CLP)" in the abstract section according to your suggestion (corresponding to lines 19-20 of the revision document). In our study, we chose the cultivated land pressure index model to represent the cultivated land pressure in the influence area of the lower Yellow River. The cultivated land pressure index (i.e., "K" in Equation (2)) refers to the ratio of the minimum per capita cultivated land area required to meet the food consumption of each person at a normal living standard to the actual per capita cultivated land area.

Comment 2: in the first part of the manuscript you used the unit "hm2". I do not know this unit and in the text I suggest to change it to "km2".

Response: Thank you for your suggestions. The term "hm2" is used in the article to indicate the unit of area in hectares, which is usually used less than "km2".
